# Extensive Structure Modification on Luteolin-Cinnamic Acid Conjugates Leading to BACE1 Inhibitors with Optimal Pharmacological Properties

**DOI:** 10.3390/molecules25010102

**Published:** 2019-12-26

**Authors:** De-Yang Sun, Chen Cheng, Katrin Moschke, Jian Huang, Wei-Shuo Fang

**Affiliations:** 1State Key Laboratory of Bioactive Substances and Functions of Natural Medicines, Institute of Materia Medica, Chinese Academy of Medical Sciences and Peking Union Medical College, 2A Nan Wei Road, Beijing 100050, China; sdy887@126.com; 2College of Life Science, Wuhan University, Wuhan 430072, Hubei, China; cccc@whu.edu.cn (C.C.); jianhuang@whu.edu.cn (J.H.); 3German Center for Neurodegenerative Diseases (DZNE) and Munich Cluster for Systems Neurology (SyNergy), Technical University of Munich, Feodor-Lynen-Strasse 17, 81377 Munich, Germany; katrin.moschke@dzne.de; 4Neuroproteomics, Klinikum rechts der Isar, Technical University of Munich, 81675 Munich, Germany

**Keywords:** beta-secretase, BACE1 inhibitor, flavonoid, luteolin, cinnamic acid

## Abstract

BACE1 inhibitory conjugates derived from two natural products, luteolin (**1**) and *p*-hydroxy-cinnamic acid (**2**), were subjected to systematic structure modifications, including various positions in luteolin segment for conjugation, different linkers (length, bond variation), as well as various substitutions in cinnamic acid segment (various substituents on benzene, and replacement of benzene by heteroaromatics and cycloalkane). Optimal conjugates such as **7c** and **7k** were chosen on the basis of a series of bioassay data for further investigation.

## 1. Introduction

Alzheimer’s disease (AD) is an age-dependent chronic neurodegenerative disease that is characterized by the presence of extracellular senile plaques and neurofibrillary tangles, as well as synaptic dysfunction and neuronal cell death. According to the World Health Organization (WHO), about 35.6 million people were diagnosed with dementia in 2010, and the number is projected to double every 20 years. The overall cost for dementia worldwide has been estimated at approximately 604 billion U.S. dollars [1].

Although the cause of AD remains in debate, β-amyloid cascade hypothesis (Aβ hypothesis) or its modified version is the most widely accepted pathological mechanism according to genetic and pathological evidence among many mechanisms. Aβ, a peptide consisting of 37–43 amino acids, is proposed as aggregating into soluble oligomers and fibrils, which are toxic to neurons, causing neuronal death. Another protein, tau, can also be a causative factor independently, presumably acting downstream of Aβ aggregation [2,3,4].

Aβ is generated by the sequential endoproteolytic cleavage of amyloid precursor protein (APP), a transmembrane protein expressed in many tissues and organs, by β-secretase (BACE1, also known as β-site amyloid precursor protein cleaving enzyme 1) and γ-secretases subsequently in the brain [5].

Thus, the key enzyme BACE1 involved in the generation of Aβ has been regarded as an attractive AD drug target for about two decades. Many different chemotypes of BACE1 inhibitors have been developed [6]. Although some BACE1 inhibitors have advanced to clinical research, none of them have finished the phase Ⅲ clinical trial due to diverse side effects and/or lack of effectiveness [7].

The failure of these clinical trials, including the most recent discontinuation of AstraZeneca/Eli Lilly’s Lanabecestat in phase III clinical trial, proposed that the early intervention of AD, even without any symptoms, may be necessary for the success of the BACE1 inhibitors [8,9]. The kickoff of prevention trial with BACE1 inhibitor CNP250 represents a new trend in BACE1 inhibitor development [10].

The side effects have arisen from, at least for some of the BACE1 inhibitor drug candidates, the indiscriminately blocking of almost all functions and substrate cleavages related to BACE1 by the active site inhibitors [11].

Unlike competitive inhibitors targeting enzyme active site, non-competitive inhibitors binding to allosteric sites may selectively inhibit specific functions of an enzyme, resulting in less side effects [12].

In our previous work [13], luteolin (**1**) and *p*-hydroxy-cinnamic acid (**2**) (Figure 1) were found to simultaneously bind different parts of BACE1 protein by STD-NMR experiment [14]. We designed the conjugates to link two segments through proper linkage to improve the potency. Conjugate **3** (Figure 1) was found in our previous work as a potent, cell-permeable, and non-competitive inhibitor. In this work, an extensive modification of **3** was designed and performed to optimize it so as to find BACE inhibitors with optimal drug-like properties.

In this paper, we first probed the different positions in flavonoid segment for conjugation of cinnamoyl segment, in order to avoid the possibly imprecise information derived from STD-NMR. After the C-7 position of flavone segment was confirmed as the optimal position among all conjugates in **Series Ⅰ**, we further designed and prepared **Series Ⅱ** and **Series Ⅲ** compounds to optimize the linker and substituents in the cinnamoyl segment of the conjugate (Figure 2).

## 2. Results and Discussion

### 2.1. Design, Synthesis, and Enzyme Inhibitory Activity of **Series Ⅰ** Compounds

Although in our previous work [13] a cell permeable BACE1 inhibitor **3** was discovered with the STD-NMR, we wished in this study to probe various positions in **1** (including 3′, 4′, 3, 6, and 8 positions) for direct conjugation of **2** or through ethylene glycol linkers, to avoid the possibly misleading of imprecise STD-NMR data. 

In general, our designed conjugates consisted of a flavone segment, a linker, and a cinnamoyl segment. However, the synthetic routes and the choice of starting material varied to a great extent for the different conjugates. 

Most luteolin analogs with hydroxyl at different positions as the flavone segment for conjugation are not commercially available, and thus were assembled from the simple starting materials aromatic aldehyde and aromatic ketone with different substituting groups, through condensation, to form chalcone structure under basic conditions, cyclized to dihydroflavone under heating conditions, and then oxidized to flavonoids with I_2_/pyridine under heating conditions (Scheme 1 and Appendix A).

For the synthesis of conjugate **4j**, the flavone segment quercertin (3-OH substituted luteolin) was prepared by de-glycosylation of a natural compound rutin (auercetin-3-rutinoside). At first, per-benzylated rutin (**14**) was subjected to acid hydrolysis due to the poor stability of rutin under acidic condition. Then, ethylene glycol linker was attached to 3-OH to afford **17**. The switch of benzyl-protecting groups in **17** to methoxymethyl group in combination of the removal of temperorary silyl protection of 3-*O*-ethylene-ol over two steps furnished **20**, which was ready for the 3-*O*-conjugation of cinnamoyl side chain. The switch of protecting group was intended to avoid the stringent deprotection conditions at the final stage of target compound preparation (Scheme 2).

The terminal hydroxyl group in **20** was esterified with 4″-acetyl cinnamic acid by activated ester method, and then the acetyl in **21** was successively removed under alkaline condition and MOM groups under acidic condition to afford desired conjugates **4j** (Scheme 3).

As seen from Table 1, the BACE1 inhibition of conjugates **4a**–**i** with *p*-hydroxy-cinnamic acid attached to C-3′ and -4′ of **1**, no matter what the linker was, and was weaker than the conjugation to other positions of **1**. Respectively, the inhibitory effect of conjugates **4j**, **4k**, and **4l** attached to C-3, C-6, and C-8 were stronger than C-3′ and -4′ conjugates but weaker than the C-7 conjugate, **3**. Conjugate **4k** was weaker than the flavone segment **1** (without conjugation of **2**). The above results demonstrated the C-7 in flavone segment as the optimal conjugation position, consistent with optimization based on STD-NMR.



In addition, the acetyl group reduced inhibitory activity for all pairs of compounds (**4a** vs. **4b**, **4c** vs. **4d**, **4g** vs. **4h**), suggesting the terminal phenol group may play a role in the enzyme inhibition. 

### 2.2. Design, Synthesis, and Activity of **Series Ⅱ** Compounds

In the previous study [13], conjugates with one to four glycols units as linkers were tested, and compound with one glycol unit (i.e., in **3**) was found to be the optimal. Here, we replaced glycol linkers by other polymethylene diols, trying to optimize the length of linker more carefully (Figure 3).

For the synthesis of **Series Ⅱ** compounds, the di-phenol groups on the C ring of **1** were protected with diphenylmethylidene to prepare **23**. The preparation of different hydroxyalkyl-7-luteolin could be realized by the treatment of **23** with TBS-*O*-hydroxyalkyl tosylate- under K_2_CO_3_/acetone, except for butylene diol, in which case the reagent 4-hydroxybutyl 1-tosylate was subjected to intramolecular reaction to furnish tetrahydrofuran under the basic condition, possibly due to high reactivity of the OTs group. Thus, 4-TBS-*O*-butyl bromide was applied to this conjugation and the key intermediate **24** successfully prepared. After the removal of silyl protective group in **24**, the terminal hydroxyl group in **25** was esterified with 4″-acetyl cinnamic acid in the presence of EDCI to furnish **26**, which was then deprotected under acidic conditions to afford desired conjugates **5a**–**f** (Scheme 4).

To explore the effect of the length of polymethylene diol linker in the conjugates, a series of conjugates **5a**–**f** with different length of linkers as well as **5g** without linker were prepared. 

In general, the inhibitory activity of conjugates **5a**–**f** decreased as the length of linker increased (Table 2). Although conjugate **5a** bearing the shortest length of linker in the series was the most active, its IC_50_ was still lower than **3** bearing ethylene glycol linker. Thus, we adhered to the ethylene glycol linker in the next round of optimization.

### 2.3. Design, Synthesis, and Activity of **Series Ⅲ** Compounds

#### 2.3.1. Design, Synthesis, and Enzyme Inhibition Activity of **Series Ⅲ** Compounds

Next, we tried to optimize the other segment (i.e., cinnamoyl segment) in the conjugate, through replacing the *p*-hydroxy by other substituents on phenyl ring, and the phenyl, by heterocycles and cyclohexane (Figure 4, Table 3 and Figure 5). In addition, the ester bond between the cinnamoyl segment and the linker was replaced by amide group to make this possibly labile bond more stable (Figure 5). 

The general synthetic routes of **Series Ⅲ** compounds were much similar to **Series Ⅱ.** It should be noted that the acyl chloride method was chosen for compound **7b**, **7d**, and **7o** when the ester bond compounds were prepared (Scheme 5).

As shown in Table 4, conjugate **7k** with pyridine in placement of phenol exhibited the most potent inhibitory activity with an IC_50_ of 0.15 μM in the series, with an approximate 10-fold enhancement compared with the flavone segment **1** or conjugate **3**. It is worth noting that many conjugates exhibited an inhibition rate of approximately 60% at 10 μM, while possessing 50% inhibition even at low concentration (e.g., **7k**, **7l** with IC_50_s of 0.1–0.2 μM). It was difficult to reach the maximum inhibition rate of 80–90% even at 20 μM or 50 μM. An S-curve for IC_50_ calculation of several compounds is shown in the Appendix A.

The structure activity relationship (SAR) for **Series III** can be summarized as follows: (1) The hydroxy on ring B of flavone segment was necessary for activity (possibly through hydrogen bonding). The inhibitory activity of conjugates decreased obviously when the hydroxy was replaced by hydrogen or other groups (**3** vs. **6**). (2) The hydroxy on the benzene ring of cinnamoyl segment was also important. The inhibitory activity of conjugates decreased when the hydroxy was replaced by other non-polar substituents, or by methylation or esterification (**3**, **7b**, **7c**, **7d** vs. **7a**, **7i**, **7q**). However, the position and number of hydroxyls did not have a significant impact on the activity (**7b**, **7c** vs. **3**). (3) The derivatives with aromatic heterocyclic showed favorable activity, especially the compound with pyridine ring. (4) When the aromatic or heteroaromatic ring (**7k**–**7l**, **7n**–**7p**) was replaced by cyclohexyl group (**7m**), the activity decreased significantly.

#### 2.3.2. Cytotoxicity of **Series Ⅲ** Compounds

It is crucial for the BACE1 inhibitors to exert their activities at sub-toxic concentration, and thus their cytotoxicity was assessed in HEK293 cells by 3-[4,5-dimethylthiazol-2-yl]-2,5-diphenyl-tetrazolium bromide (MTT) assays (Table 5). Among the compounds, **7a**, **7h**, **7q**, **7k**, and **8** were the least cytotoxic (IC_50_ > 100 μM), and **7c** and **7p** were weak cytotoxic (IC_50_ = 90 µM). Unfortunately, there was no obvious relationship between the structures and their cytotoxicity.

#### 2.3.3. Enzyme Inhibition in Cells

To assess the cellular BACE1 inhibitory activity of the conjugates, **7a**–**d**, **7i**, and **7k** were incubated in APP overexpressing HEK293 cells and their inhibition on the three main Aβ species Aβ-38, Aβ-40, and Aβ-42—the main amyloidogenic peptides produced by BACE1 cleavage of APP peptide—was measured by an ELISA-like immunoassay method. Aβ levels were normalized to the protein concentration in the lysate in order to rule out differences in total cell number per well of the culture plate. Additionally, the normalized Aβ level was set to 100% for the DMSO-treated control.

The Aβ levels upon treatment of different compounds are shown in Figure 6 (Aβ-42) and the Appendix A (Aβ-38 and Aβ-40). At the higher concentrations (10, 25, and 50 µM), **7b**–**d** and **1** dose-dependently reduced Aβ levels. It is worth noting that **7c** showed an obvious inhibition Aβ-42 level at 25 µM. However, **7i** and **7k** did not reduce Aβ levels, and even increased Aβ-42 levels.

#### 2.3.4. Enzyme Kinetics Experiment and Enzyme Selectivity

We are interested to know if the most active conjugates derived from **1** and **3** will keep or change their non-competitive mode of inhibition. The Dixon plot was used to determine the kinetics of **4l**, **7c**, and **7k** inhibiting BACE1. As shown in Figure 7, the three lines that stand for different substrate concentration intersect on the *x*-axis, demonstrating that **7c** was a non-competitive inhibitor with a Ki value of 1.2 µM, and that **4l** and **7k** were in mix-competitive mode (Appendix A).

As BACE1 is a member of aspartyl protease family, its selective inhibition over other closely related proteases is required to avoid possible severe side effects. We measured the inhibition of **7c**, **7k**, and **8** on BACE-2 and renin, and found **7c** and **7k** showed very good selectivity (ca. 2–3 orders of magnitude) on BACE1 over BACE2 and renin (Table 6).

To assess the potential of these conjugates as CNS drug candidates, the ability to cross the BBB (blood-brain barrier) is worthy evaluation. Thus, we calculated cLogP and BBB penetration for several representative conjugates by an open source software [16]. The result (Appendix A) showed that some of them (e.g., **7k**) may have the ability to cross the BBB.

In summary, the extensive optimization of flavone-cinnamic acid conjugates revealed some more potent BACE1 inhibitors including **7c** and **7k**. On the basis of the analysis of multiple bioassay data, **7c** exhibited improved activity and weaker toxicity in cells, demonstrating an improvement from the starting point of this work, conjugate **3**. Moreover, **7c** also showed non-competitive mode of BACE1 inhibition and selectivity over other related enzymes such as BACE2 and renin, suggesting it may retain the selective inhibition of BACE1 as an allosteric inhibitor to facilitate the reduction of APP processing and spare other BACE1 functions. Combining the computational result of LogP and BBB-penetrating ability, it was demonstrated that such a conjugate may be served as a drug candidate for AD therapy, and in-depth PD (pharmacodynamic) and PK (pharmacokinetic) studies in animals are warranted. 

## 3. Materials and Methods

### 3.1. Chemical Synthesis

In this part, the preparation for representative conjugates are shown. Other compounds and also the NMR and mass spectrometry (MS) data for structural characterization for all compounds are included in the Appendix A.

#### 3.1.1. General Methods

All chemicals and reagents were purchased from Beijing Innochem Science and Technology Co. Ltd. (Beijing, China), Sinopharm Chemical Reagent Co. Ltd. (Beijing, China), and thin-layer chromatography (TCI). The 200–300 mesh silica gel used for flash column chromatography was purchased from Rushanshi Shuangbang Xincailiao Co. Ltd. (Rushan, Shandong, China). Visualization on TLC (analytical thin layer chromatography) was achieved by the use of UV light (254 and 365 nm). All solvents were purified and dried according to the standard procedures. The purification was performed on flash column chromatography. The high performance liquid chromatography (HPLC)-electrospray ionization (ESI)-mass spectrometry (MS) analysis was carried out in an Agilent 1260 Infinity HPLC system (Agilent Technologies, Waldbronn, Germany) equipped with a reversed phase 4.68 × 50 mm (1.8 µm) XDB-C18 Column and consisted of a binary solvent delivery system, an auto sampler, a column temperature controller, and a UV detector. The mass spectra were acquired by a 6120 Quadrupole LC-MS mass spectrometer (Agilent Technologies, Waldbronn, Germany) connected to the HPLC system via an ESI interface. Proton and carbon magnetic resonance spectra (^1^H-NMR and ^13^C-NMR) were recorded on a Bruker BioSpin AG 300 or 400 MHz spectrometer or Varian 400, 500, or 600 MHz spectrometer. ^1^H-NMR data were reported as follows: chemical shifts, multiplicity (s = singlet, d = doublet, t = triplet, q = quartet, m = multiplet), coupling constant(s) in Hz, integration. All tested compounds **4a**–**l**, **5a**–**g**, **6**, **7a**–**q**, and **8** were ≥95% pure by HPLC (column C18 4.6 × 250 mm 5 µm, mobile phase: acetonitrile–water (70:30 or 80:20 or 90:10 in 10 min), flow rate 1.0 mL/min, detected at 254 nm). 

#### 3.1.2. Synthesis of Compound **4j**

*7-(benzyloxy)-2-(3,4-bis(benzyloxy)phenyl)-3,5-dihydroxy-4H-chromen-4-one* (**16**): To a solution of rutin (610 mg, 1 mmol) in DMF (30 mL), BnBr (684 mg, 4 mmol) and K_2_CO_3_ (552 mg, 4 mmol) were added. The solution was stirred at 50 °C for 3 h. Water (100 mL) was added, and the solution was cooled to 0 °C for overnight. The solid **15** was collected by suction filtration and dissolved in 95% alcohol (20 mL), and hydrochloric acid (3 mL) was added. After stirring for 2 h at 70 °C, solid was precipitated in the solution. Then, compound **16** (740 mg, 84%) was collected by suction filtration after the solution was cooled to r.t.

*7-(benzyloxy)-2-(3,4-bis(benzyloxy)phenyl)-3-(2-((tert-butyldimethylsilyl)oxy)ethoxy)-5-hydroxy-4H-chromen-4-one* (**17**): To a solution of **16** (147 mg, 0.26 mmol) in DMF (5 mL), K_2_CO_3_ (54 mg, 0.39 mmol) and (2-bromoethoxy)(*tert*-butyl)dimethylsilane (93 mg, 0.39mmol) were added. After stirring at 80 °C for 12 h, water (20 mL) was added and extracted with ethyl acetate (25 mL × 2). The organic layer was gathered, dried over Na_2_SO_4_, filtered, and concentrated in vacuo. The crude product was purified by column chromatography (PE/acetone = 10:1) to give **17** as yellow solids (101 mg, 52.7%).

*3-(2-((tert-butyldimethylsilyl)oxy)ethoxy)-2-(3,4-dihydroxyphenyl)-5,7-dihydroxy-4H-chromen-4-one* (**18**): To a mixture of **17** (100 mg, 0.14 mmol) and Pd/C (150 mg) in THF (5 mL) and ethanol (5 mL), 1,4-cyclohexadiene (659 μL, 7 mmol) was added. After stirring at 80 °C for 12 h, the solution was concentrated in vacuo after suction filtration. The crude product was purified by column chromatography (PE/acetone/THF = 9:2:1) to give **18** as yellow solids (49 mg, 78.6%). 

*2-(3,4-bis(methoxymethoxy)phenyl)-3-(2-((tert-butyldimethylsilyl)oxy)ethoxy)-5-hydroxy-7-(methoxymethoxy)-4H-chromen-4-one* (**19**): The starting material 18 (310 mg, 0.67 mmol) was dissolved in DCM (5 mL), and MOMCl (0.18 mL, 2.16 mmol) was added at −10 °C. After 10 mins stirring, DIEA (0.44 mL, 2.16 mmol) was added slowly and the suspension heated to 0 °C. Saturated sodium bicarbonate (10 mL) was added to quench reaction after 1 h. The dichloromethane layer was separated, then the aqueous phase was extracted with dichloromethane (10 mL). The organic phase was merged, then washed with water and brine, dried over Na_2_SO_4_, filtered, and concentrated in vacuo. The residue was purified by column chromatography (petroleum ether/acetone = 10:1) to give **19** as yellow solids (230 mg, 57.4%). 

*2-(3,4-bis(methoxymethoxy)phenyl)-5-hydroxy-3-(2-hydroxyethoxy)-7-(methoxymethoxy)-4H-chromen-4-one* (**20**): A solution of compound 19 (230 mg, 0.39 mmol) and CsF (304 mg, 2 mmol) in dry DMF (2 mL) was heated to 100 °C for 2 h. The resulting mixture was diluted with water (5 mL) and extracted with ethyl acetate (10 mL × 3). The organic layer was dried over Na_2_SO_4_, filtered, and concentrated in vacuo. The crude product was purified by column chromatography (petroleum ether/acetone = 5:1) to give **20** as yellow solid (154 mg, 83.0%).

*2-((2-(3,4-bis(methoxymethoxy)phenyl)-5-hydroxy-7-(methoxymethoxy)-4-oxo-4H-chromen-3-yl)oxy)ethyl (E)-3-(4-acetoxyphenyl)acrylate* (**21**): To a solution of 4-acetoxycinnamic acid (41 mg, 0.2 mmol) in THF (1 mL), EDCI (38 mg, 0.2 mmol), DMAP (6 mg, 0.05 mmol), and Et_3_N (45 μL, 0.3 mmol) were added. Then, **16a** (27 mg, 0.066 mmol) in THF (1 mL) was added under nitrogen. The solution was stirred at r.t. for 12 h. The organic layer was concentrated in vacuo. The residue was purified by column chromatography (petroleum ether/EtOAc = 3:1) to give **21** as white solid (24 mg, 45.0%).

*2-((2-(3,4-bis(methoxymethoxy)phenyl)-5-hydroxy-7-(methoxymethoxy)-4-oxo-4H-chromen-3-yl)oxy)ethyl (E)-3-(4-hydroxyphenyl)acrylate* (**22**): Compound **21** (24 mg, 0.036 mmol) was dissolved in a solution of CH_2_Cl_2_ (1 mL) and a K_2_CO_3_ (1.38 mg, 0.01 mmol) MeOH solution (1 mL) was added under argon at room temperature. After stirring for 12 h, the resulting mixture was concentrated in vacuo and purified by column chromatography (DCM/acetone = 30:1) to give **22** as yellow solid (12 mg, 56.2%).

*2-((2-(3,4-dihydroxyphenyl)-5,7-dihydroxy-4-oxo-4H-chromen-3-yl)oxy)ethyl(E)-3-(4-hydroxyphenyl)acrylate* (**4j**): To a solution of **22** (12 mg, 0.019 mmol) in 0.2 mL acetone, 38% HCl/AcOH (*v/v* = 1:15, 1.8 mL) mixture was added at room temperature. The solution was stirred at room temperature. After 24 h, yellow solid precipitated and the precipitate was filtered off, washed with water, and dried to give **4j** (5.1 mg, 53.5%) as yellow solid. 

#### 3.1.3. Synthesis of Compound **5a**–**f**

*Luteolin 3’,4’-O-diphenylmethylidene* (**23**): Dichlorodiphenylmethane (20 μL, 1.1 mmol) was added to a stirred mixture of luteolin (20 mg, 0.07 mmol) in diphenyl ether (1.4 mL), and the reaction mixture was heated at 165 °C for 2.5 h. After being cooled to room temperature, the reaction solution was poured into petroleum ether (20 mL), and the precipitation was filtered and washed with petroleum ether. The filter residue was dissolved into acetone, and the resulting solution was concentrated and purified by column chromatography (petroleum ether/EtOAc = 4:1) to give **45** as yellow solid (22 mg, 70%). 

*7-(4-((tert-butyldimethylsilyl)oxy)butoxy)-2-(2,2-diphenylbenzo[d][1,3]dioxol-5-yl)-5-hydroxy-4H-chromen-4-one* (**24**): TBSCl (452 mg, 3.0 mmol) and imidazole (204 mg, 3.0 mmol) were added to a solution of 4- bromobutanol (300 mg, 2.0 mmol) in DMF (3 mL) at room temperature. After stirring overnight, the resulting mixture was poured into ice-water, and extracted with EtOAc. The organic layer was washed with saturated aqueous NaHCO_3_ and NaCl successively, dried over Na_2_SO_4_, filtered, and concentrated in vacuo to give Br(CH_2_)_4_OTBS crude. The crude was dissolved in acetone (3 mL), then **23** (225 mg, 0.5 mmol) and K_2_CO_3_ (276 mg, 2 mmol) were added. The reaction mixture was then refluxed overnight and concentrated in vacuo. The residue was purified by column chromatography (petroleum ether/acetone = 7:1) to give **24** as yellow solid (136 mg, 42.8%). 

*2-(2,2-diphenylbenzo[d][1,3]dioxol-5-yl)-5-hydroxy-7-(3-hydroxypropoxy)-4H-chromen-4-one* (**25a**): Potassium carbonate (143 mg, 1.0 mmol) was added to a solution of **23** (315 mg, 0.7 mmol) and TsO(CH_2_)_3_OH (230 mg, 1.0 mmol) in CH_3_CN (4 mL) at room temperature. The reaction mixture was then refluxed overnight. The resulting mixture was poured into ice-water, adjusted to neutral pH with aqueous HCl (5%), and extracted with EtOAc. The organic layer was washed with brine, dried over Na_2_SO_4_, filtered, and concentrated in vacuo. The residue was purified by column chromatography (petroleum ether/acetone = 5:1) to give **25a** as yellow solid (218 mg, 34.2%). 

*2-(2,2-diphenylbenzo[d][1,3]dioxol-5-yl)-5-hydroxy-7-((5-hydroxypentyl)oxy)-4H-chromen-4-one* (**25c**): Compound **25c** (60 mg, 27.8%) was synthesized from TsO(CH_2_)_5_OH (104 mg, 0.4 mmol), according to the procedure used to prepare **25a**, obtained as yellow solid. 

*2-(2,2-diphenylbenzo[d][1,3]dioxol-5-yl)-5-hydroxy-7-((6-hydroxyhexyl)oxy)-4H-chromen-4-one* (**25d**): Compound **25d** (132 mg, 47.5%) was synthesized from TsO(CH_2_)_6_OH (133 mg, 0.49 mmol), according to the procedure used to prepare **25a**, obtained as yellow solid. 

*2-(2,2-diphenylbenzo[d][1,3]dioxol-5-yl)-5-hydroxy-7-((8-hydroxyoctyl)oxy)-4H-chromen-4-one* (**25e**): Compound **25e** (237 mg, 55.3 %) was synthesized from TsO(CH_2_)_8_OH (288 mg, 0.96 mmol), according to the procedure used to prepare **25a**, obtained as yellow solid. 

*2-(2,2-diphenylbenzo[d][1,3]dioxol-5-yl)-5-hydroxy-7-((10-hydroxydecyl)oxy)-4H-chromen-4-one* (**25f**): Compound **25f** (212 mg, 49.9%) was synthesized from TsO(CH_2_)_10_OH (340 mg, 1.04 mmol), according to the procedure used to prepare **25a**, obtained as yellow solid. 

*2-(2,2-diphenylbenzo[d][1,3]dioxol-5-yl)-5-hydroxy-7-(4-hydroxybutoxy)-4H-chromen-4-one* (**25b**): HF (743 μL, 17.1 mmol) and pyridine (1.4 mL, 17.1 mmol) were added to a solution of **24** (36 mg, 0.057 mmol) in MeCN (1 mL) at room temperature. After stirring overnight, the reaction was quenched by 1N hydrochloric acid, and extracted with EtOAc. The organic layer was washed with saturated aqueous NaHCO_3_ and NaCl successively, dried over Na_2_SO_4_, filtered, and concentrated in vacuo to give **25b** (24 mg, 80.7%) as yellow solid.

*(E)-3-((2-(2,2-diphenylbenzo[d][1,3]dioxol-5-yl)-5-hydroxy-4-oxo-4H-chromen-7-yl)oxy)propyl 3-(4-acetoxyphenyl)acrylate* (**26a**): Compound **26a** (34 mg, 24.4%) was synthesized from **25a** (101 mg, 0.2 mmol) according to the procedure used to prepare **17**. The crude product was purified by column chromatography (PE/acetone = 5:1) to give **26a** as yellow solids.

*(E)-4-((2-(2,2-diphenylbenzo[d][1,3]dioxol-5-yl)-5-hydroxy-4-oxo-4H-chromen-7-yl)oxy)butyl 3-(4-acetoxyphenyl)acrylate* (**26b**): Compound **26b** (19 mg, 58.1%) was synthesized from **25b** (24 mg, 0.046 mmol) according to the procedure used to prepare **17**. The crude product was purified by column chromatography (PE/DCM/acetone = 10:10:1) to give **25b** as yellow solids.

*(E)-5-((2-(2,2-diphenylbenzo[d][1,3]dioxol-5-yl)-5-hydroxy-4-oxo-4H-chromen-7-yl)oxy)pentyl 3-(4-acetoxyphenyl)acrylate* (**26c**): Compound **26c** (17 mg, 36.1%) was synthesized from **25c** (35 mg, 0.065 mmol) according to the procedure used to prepare **17**. The crude product was purified by column chromatography (PE/DCM/acetone = 10:10:1) to give **26c** as yellow solids.

*(E)-6-((2-(2,2-diphenylbenzo[d][1,3]dioxol-5-yl)-5-hydroxy-4-oxo-4H-chromen-7-yl)oxy)hexyl 3-(4-acetoxyphenyl)acrylate* (**26d**): Compound **26d** (35 mg, 33.8%) was synthesized from **25d** (75 mg, 0.14 mmol) according to the procedure used to prepare **17**. The crude product was purified by column chromatography (PE/DCM/acetone = 10:10:1) to give **26d** as yellow solids. 

*(E)-8-((2-(2,2-diphenylbenzo[d][1,3]dioxol-5-yl)-5-hydroxy-4-oxo-4H-chromen-7-yl)oxy)octyl 3-(4-acetoxyphenyl)acrylate* (**26e**): Compound **26e** (52 mg, 40.1%) was synthesized from **25e** (100 mg, 0.17 mmol) according to the procedure used to prepare **17**. The crude product was purified by column chromatography (PE/DCM/THF = 20:20:3) to give **26e** as yellow solids. 

*(E)-10-((2-(2,2-diphenylbenzo[d][1,3]dioxol-5-yl)-5-hydroxy-4-oxo-4H-chromen-7-yl)oxy)decyl 3-(4-acetoxyphenyl)acrylate (***26f**): Compound **26f** (44 mg, 30.7%) was synthesized from **25f** (110 mg, 0.18 mmol) according to the procedure used to prepare **17**. The crude product was purified by column chromatography (PE/DCM/THF = 20:20:3) to give **26f** as yellow solids.

*(E)-3-((2-(3,4-dihydroxyphenyl)-5-hydroxy-4-oxo-4H-chromen-7-yl)oxy)propyl 3-(4-hydroxyphenyl)acrylate* (**5a**): Compound **5a** (12 mg, 48.9%) was synthesized from **26a** (34 mg, 0.05 mmol) according to the procedure used to prepare **4j**. The crude product was purified by column chromatography (chloroform/methanol = 35:1) to give **5a** as yellow solids.

*(E)-4-((2-(3,4-dihydroxyphenyl)-5-hydroxy-4-oxo-4H-chromen-7-yl)oxy)butyl 3-(4-hydroxyphenyl)acrylate* (**5b**): Compound **5b** (5.2 mg, 36.8%) was synthesized from **26b** (19 mg, 0.028 mmol), according to the procedure used to prepare **5a,** to give **5b** as yellow solids.

*(E)-5-((2-(3,4-dihydroxyphenyl)-5-hydroxy-4-oxo-4H-chromen-7-yl)oxy)pentyl 3-(4-hydroxyphenyl)acrylate***,** (**5c**): Compound **5c** (5.1 mg, 42.8%) was synthesized from **26c** (17 mg, 0.023 mmol), according to the procedure used to prepare **5a,** to give **5c** as yellow solids.

*(E)-6-((2-(3,4-dihydroxyphenyl)-5-hydroxy-4-oxo-4H-chromen-7-yl)oxy)hexyl 3-(4-hydroxyphenyl)acrylate* (**5d**): Compound **5d** (6.9 mg, 27.6%) was synthesized from **26d** (35 mg, 0.047 mmol), according to the procedure used to prepare **5a,** to give **5d** as yellow solids.

*(E)-8-((2-(3,4-dihydroxyphenyl)-5-hydroxy-4-oxo-4H-chromen-7-yl)oxy)octyl 3-(4-hydroxyphenyl)acrylate* (**5e**): Compound **5e** (8.2 mg, 29.3%) was synthesized from **26e** (38 mg, 0.05 mmol), according to the procedure used to prepare **5a,** to give **5e** as yellow solids. 

*(E)-10-((2-(3,4-dihydroxyphenyl)-5-hydroxy-4-oxo-4H-chromen-7-yl)oxy)decyl 3-(4-hydroxyphenyl)acrylate* (**5f**): Compound **5f** (7.3 mg, 27.6%) was synthesized from **26f** (36 mg, 0.045 mmol), according to the procedure used to prepare **5a,** to give **5f** as yellow solids. 

#### 3.1.4. Synthesis of Compound **7a**–**q**

*2-(2,2-diphenylbenzo[d][1,3]dioxol-5-yl)-5-hydroxy-7-(2-hydroxyethoxy)-4H-chromen-4-one* (**27**): Compound **27** (195 mg, 57%) was synthesized from **23** according to the procedure used to prepare **25a**. The crude product was purified by column chromatography (CH_2_Cl_2_/acetone = 10:1) to give **27** as yellow solids.

*2-((2-(2,2-diphenylbenzo[d][1,3]dioxol-5-yl)-5-hydroxy-4-oxo-4H-chromen-7-yl)oxy)ethyl cinnamate* (**28a**): Compound **28a** (28 mg, 57.6%) was synthesized from **27** (40 mg, 0.08 mmol) and cinnamic acid (35 mg, 0.24 mmol) according to the procedure used to prepare **21**. The crude product was purified by column chromatography (PE/acetone = 5:1) to give **28a** as yellow solids.

*(E)-2-((2-(2,2-diphenylbenzo[d][1,3]dioxol-5-yl)-5-hydroxy-4-oxo-4H-chromen-7-yl)oxy)ethyl 3-(2-acetoxyphenyl)acrylate* (**28b**): To a solution of 2-acetoxycinnamic acid (54 mg, 0.26 mmol) in dichloroethane (5 mL), SOCl_2_ (100 μL, 2.6 mmol) and DMF (1d) were added. The solution was stirred at 80 °C for 1 h, the organic solvent was removed under vacuum, and the residue dissolved in dry CH_2_Cl_2_ (4 mL). Pyridine (31 μL, 0.38 mmol) was added at 0 °C, then **27** (64 mg, 0.13 mmol) was added dropwise. The resulting mixture was transferred to r.t. and stirred for 4 h. The organic solvent was washed with 1M hydrochloric acid and brine, dried over Na_2_SO_4_, filtered, and concentrated in vacuo. The crude product was purified by column chromatography (PE/acetone = 5:1) to give **28b** as yellow solid (58 mg, 50.3%).

*(E)-2-((2-(2,2-diphenylbenzo[d][1,3]dioxol-5-yl)-5-hydroxy-4-oxo-4H-chromen-7-yl)oxy)ethyl 3-(3-acetoxyphenyl)acrylate* (**28c**): Compound **28c** (48 mg, 35.2%) was synthesized from **27** (100 mg, 0.2 mmol) and 3-acetoxycinnamic acid (124 mg, 0.6 mmol), according to the procedure used to prepare **21**. The crude product was purified by column chromatography (PE/acetone = 5:1) to give **28c** as yellow solids.

*(E)-4-(3-(2-((2-(2,2-diphenylbenzo[d][1,3]dioxol-5-yl)-5-hydroxy-4-oxo-4H-chromen-7-yl)oxy)ethoxy)-3-oxoprop-1-en-1-yl)-1,2-phenylene diacetate* (**28d**): Compound **28d** (24 mg, 46.3%) was synthesized from **27** (35 mg, 0.07 mmol) and 3,4-diacetoxycinnamic acid (28 mg, 0.11 mmol), according to the procedure used to prepare **28b**. The crude product was purified by column chromatography (PE/acetone = 4:1) to give **28d** as yellow solids. 

*(E)-2-((2-(2,2-diphenylbenzo[d][1,3]dioxol-5-yl)-5-hydroxy-4-oxo-4H-chromen-7-yl)oxy)ethyl 3-(4-fluorophenyl)acrylate* (**28e**): Compound **28e** (37 mg, 44.3%) was synthesized from **27** (64 mg, 0.13 mmol) and 4-fluorocinnamic acid (65 mg, 0.39 mmol), according to the procedure used to prepare **21**. The crude product was purified by column chromatography (PE/acetone = 5:1) to give **28e** as white solids.

*(E)-2-((2-(2,2-diphenylbenzo[d][1,3]dioxol-5-yl)-5-hydroxy-4-oxo-4H-chromen-7-yl)oxy)ethyl 3-(4-chlorophenyl)acrylate* (**28f**): Compound **28f** (44 mg, 55.6%) was synthesized from **27** (60 mg, 0.12 mmol) and 4-chlorocinnamic acid (66 mg, 0.36 mmol), according to the procedure used to prepare **21**. The crude product was purified by column chromatography (PE/acetone = 5:1) to give **28f** as yellow solids. 

*(E)-2-((2-(2,2-diphenylbenzo[d][1,3]dioxol-5-yl)-5-hydroxy-4-oxo-4H-chromen-7-yl)oxy)ethyl 3-(4-nitrophenyl)acrylate* (**28g**): Compound **28g** (34 mg, 55.8%) was synthesized from **27** (45 mg, 0.09mmol) and 4-nitrocinnamic acid (52 mg, 0.27 mmol), according to the procedure used to prepare **21**. The crude product was purified by column chromatography (PE/acetone = 5:1) to give **28g** as white solids.

*(E)-2-((2-(2,2-diphenylbenzo[d][1,3]dioxol-5-yl)-5-hydroxy-4-oxo-4H-chromen-7-yl)oxy)ethyl 3-(4-(dimethylamino)phenyl)acrylate* (**28h**): Compound **28h** (24 mg, 45.1%) was synthesized from **27** (40 mg, 0.08 mmol) and 4-dimethylaminocinnamic acid (46 mg, 0.24 mmol), according to the procedure used to prepare **21**. The crude product was purified by column chromatography (PE/acetone = 5:1) to give **28h** as white solids.

*(E)-2-((2-(2,2-diphenylbenzo[d][1,3]dioxol-5-yl)-5-hydroxy-4-oxo-4H-chromen-7-yl)oxy)ethyl 3-(4-methoxyphenyl)acrylate* (**28i**): Compound **28i** (44 mg, 42.1%) was synthesized from **27** (80 mg, 0.16 mmol) and 4-methoxycinnamic acid (86 mg, 0.48 mmol), according to the procedure used to prepare **21**. The crude product was purified by column chromatography (PE/acetone = 5:1) to give **28i** as yellow solids.

*(E)-2-((2-(2,2-diphenylbenzo[d][1,3]dioxol-5-yl)-5-hydroxy-4-oxo-4H-chromen-7-yl)oxy)ethyl 3-(4-(tert-butyl)phenyl)acrylate* (**28j**): Compound **28j** (31 mg, 48.4%) was synthesized from **27** (51 mg, 0.1 mmol) and 4-*tert*-butylcinnamic acid (61 mg, 0.3 mmol), according to the procedure used to prepare **21**. The crude product was purified by column chromatography (PE/acetone = 5:1) to give **28j** as white solids.

*(E)-2-((2-(2,2-diphenylbenzo[d][1,3]dioxol-5-yl)-5-hydroxy-4-oxo-4H-chromen-7-yl)oxy)ethyl 3-(pyridin-3-yl)acrylate* (**28k**): Compound **28k** (63 mg, 63.0%) was synthesized from **27** (80 mg, 0.16 mmol) and (*E*)-3-(pyridin-3-yl)acrylic acid (46 mg, 0.48mmol), according to the procedure used to prepare **21**. The crude product was purified by column chromatography (PE/acetone = 3:1) to give **28k** as yellow solids.

*(E)-2-((2-(2,2-diphenylbenzo[d][1,3]dioxol-5-yl)-5-hydroxy-4-oxo-4H-chromen-7-yl)oxy)ethyl 3-(furan-3-yl)acrylate* (**28l**): Compound **28l** (36 mg, 36.6%) was synthesized from **27** (80 mg, 0.16 mmol) and (*E*)-3-(furan-3-yl)acrylic acid (66 mg, 0.48 mmol), according to the procedure used to prepare **21**. The crude product was purified by column chromatography (PE/acetone = 5:1) to give **28l** as white solids.

*(E)-2-((2-(2,2-diphenylbenzo[d][1,3]dioxol-5-yl)-5-hydroxy-4-oxo-4H-chromen-7-yl)oxy)ethyl 3-cyclohexylacrylate* (**28m**): Compound **28m** (43 mg, 42.7%) was synthesized from **27** (80 mg, 0.16 mmol) and (*E*)-3-cyclohexylacrylic acid (74 mg, 0.48 mmol), according to the procedure used to prepare **21**. The crude product was purified by column chromatography (PE/DCM/acetone = 20:20:1) to give **28m** as yellow solids.

*(E)-2-((2-(2,2-diphenylbenzo[d][1,3]dioxol-5-yl)-5-hydroxy-4-oxo-4H-chromen-7-yl)oxy)ethyl 3-(3,5-dimethyl-1H-pyrazol-1-yl)acrylate* (**28n**): Compound **28n** (42 mg, 81.6%) was synthesized from **27** (41 mg, 0.08 mmol) and 3-(3,5-dimethyl-1H-pyrazol-1-yl)propanoic acid (41 mg, 0.24 mmol), according to the procedure used to prepare **21**. The crude product was purified by column chromatography (PE/acetone = 4:1) to give **28m** as yellow solids. 

*2-((2-(2,2-diphenylbenzo[d][1,3]dioxol-5-yl)-5-hydroxy-4-oxo-4H-chromen-7-yl)oxy)ethyl 4-acetoxybenzoate* (**28o**): Compound **28o** (24 mg, 53.7%) was synthesized from **27** (31 mg, 0.07 mmol) and 4-hydroxybenzoic acid (29 mg, 0.21 mmol), according to the procedure used to prepare **28b**. The crude product was purified by column chromatography (PE/acetone = 5:1) to give **28o** as yellow solids.

*(E)-2-(2-(2,2-diphenylbenzo[d][1,3]dioxol-5-yl)-5-hydroxy-4-oxo-4H-chromen-7-yloxy)ethyl 3-(4-acetoxyphenyl)acrylate* (**28q**): Compound **28q** (76 mg, 55.7%) was synthesized from **27** (100 mg, 0.2 mmol) and 4-acetoxycinnamic acid (124 mg, 0.6 mmol), according to the procedure used to prepare **21**. The crude product was purified by column chromatography (PE/acetone = 5:1) to give **28q** as yellow solids.

*2-((2-(3,4-dihydroxyphenyl)-5-hydroxy-4-oxo-4H-chromen-7-yl)oxy)ethyl cinnamate* (**7a**): **28a** (9 mg, 0.015 mmol) was dissolved in a solution of acetone (0.1 mL) and 38% HCl/AcOH (*v/v* = 1:15, 0.5 mL). The mixture was stirred at room temperature for 20 h. The resulting mixture was poured into ice-water and extracted with EtOAc. The organic layer was washed with brine, dried over anhydrous Na_2_SO_4_, and concentrated in vacuo. The residue was purified by column chromatography (DCM/MeOH = 50:1) to give **7a** as yellow solid (2.4 mg, 32.6%). 

*(E)-2-((2-(3,4-dihydroxyphenyl)-5-hydroxy-4-oxo-4H-chromen-7-yl)oxy)ethyl 3-(2-acetoxyphenyl)acrylate* (**7b**): Compound **7b** (26.7 mg, 75.8%) was synthesized from **28b** (49 mg, 0.072 mmol) according to the procedure used to prepare **7a**, obtained as brown solids.

*(E)-2-((2-(3,4-dihydroxyphenyl)-5-hydroxy-4-oxo-4H-chromen-7-yl)oxy)ethyl 3-(3-acetoxyphenyl)acrylate* (**7c**): Compound **7c** (12.8 mg, 45.6%) was synthesized from **28c** (40 mg, 0.059 mmol) according to the procedure used to prepare **7a**, obtained as yellow solids.

*(E)-4-(3-(2-((2-(3,4-dihydroxyphenyl)-5-hydroxy-4-oxo-4H-chromen-7-yl)oxy)ethoxy)-3-oxoprop-1-en-1-yl)-1,2-phenylene diacetate* (**7d**): Compound **7d** (2.2 mg, 37.3%) was synthesized from **28d** (6 mg, 0.012 mmol), according to the procedure used to prepare **7a**. The crude product was purified by column chromatography (DCM/MeOH = 20:1) to give **7d** as white solids. 

*(E)-2-((2-(3,4-dihydroxyphenyl)-5-hydroxy-4-oxo-4H-chromen-7-yl)oxy)ethyl 3-(4-fluorophenyl)acrylate* (**7e**): Compound **7e** (9.3 mg, 62.7%) was synthesized from **28e** (23 mg, 0.031 mmol) according to the procedure used to prepare **7a**, obtained as light-yellow solids.

*(E)-2-((2-(3,4-dihydroxyphenyl)-5-hydroxy-4-oxo-4H-chromen-7-yl)oxy)ethyl 3-(4-chlorophenyl)acrylate* (**7f**): Compound **7f** (26 mg, 87.7%) was synthesized from **28f** (40 mg, 0.06 mmol) according to the procedure used to prepare **7a** without post process, obtained as white solids. 

*(E)-2-((2-(3,4-dihydroxyphenyl)-5-hydroxy-4-oxo-4H-chromen-7-yl)oxy)ethyl 3-(4-nitrophenyl)acrylate* (**7g**): Compound **7g** (4.1 mg, 63.5%) was synthesized from **28g** (9 mg, 0.013 mmol) according to the procedure used to prepare **7a**, obtained as white solids.

*(E)-2-((2-(3,4-dihydroxyphenyl)-5-hydroxy-4-oxo-4H-chromen-7-yl)oxy)ethyl 3-(4-(dimethylamino)phenyl)acrylate* (**7h**): Compound **7h** (5.1 mg, 49.7%) was synthesized from **28h** (15 mg, 0.02 mmol) according to the procedure used to prepare **7a**, obtained as yellow solids.

*(E)-2-((2-(3,4-dihydroxyphenyl)-5-hydroxy-4-oxo-4H-chromen-7-yl)oxy)ethyl 3-(4-methoxyphenyl)acrylate* (**7i**): Compound **7i** (18.8 mg, 57.3%) was synthesized from **28i** (44 mg, 0.067 mmol) according to the procedure used to prepare **7a**, obtained as white solids

*(E)-2-((2-(3,4-dihydroxyphenyl)-5-hydroxy-4-oxo-4H-chromen-7-yl)oxy)ethyl 3-(4-(tert-butyl)phenyl)acrylate* (**7j**): Compound **7j** (8.8 mg, 46.3%) was synthesized from **28j** (25 mg, 0.037 mmol) according to the procedure used to prepare **7a**, obtained as white solids.

*(E)-2-((2-(3,4-dihydroxyphenyl)-5-hydroxy-4-oxo-4H-chromen-7-yl)oxy)ethyl 3-(pyridin-3-yl)acrylate* (**7k**): Compound **7k** (12 mg, 57.3%) was synthesized from **28k** (20 mg, 0.067 mmol) according to the procedure used to prepare **7a**, obtained as yellow solids.

*(E)-2-((2-(3,4-dihydroxyphenyl)-5-hydroxy-4-oxo-4H-chromen-7-yl)oxy)ethyl 3-(furan-3-yl)acrylate* (**7l**): Compound **7l** (4.4 mg, 42.5%) was synthesized from **28l** (14 mg, 0.023 mmol) according to the procedure used to prepare **7a**, obtained as light-yellow solids.

*(E)-2-((2-(3,4-dihydroxyphenyl)-5-hydroxy-4-oxo-4H-chromen-7-yl)oxy)ethyl 3-cyclohexylacrylate* (**7m**): Compound **7m** (19.2 mg, 60.6%) was synthesized from **28m** (43 mg, 0.068 mmol) according to the procedure used to prepare **7a**, obtained as white solids.

*(E)-2-((2-(3,4-dihydroxyphenyl)-5-hydroxy-4-oxo-4H-chromen-7-yl)oxy)ethyl 3-(3,5-dimethyl-1H-pyrazol-1-yl)acrylate* (**7n**): Compound **7n** (14.2 mg, 59.2%) was synthesized from **28n** (32 mg, 0.05 mmol) according to the procedure used to prepare **7a**, obtained as yellow solids.

*2-((2-(3,4-dihydroxyphenyl)-5-hydroxy-4-oxo-4H-chromen-7-yl)oxy)ethyl 4-hydroxybenzoate* (**7o**): Compound **7o** (5.2 mg, 57.8%) was synthesized from **28o** (12 mg, 0.02 mmol) according to the procedure used to prepare **7a**, obtained as yellow solids.

*2-((2-(3,4-dihydroxyphenyl)-5-hydroxy-4-oxo-4H-chromen-7-yl)oxy)ethyl 3-(4-hydroxyphenyl)propanoate* (**7p**): To a solution of compound **28p** (24 mg, 0.038 mmol) in THF (1 mL) and EtOH (1 mL), 10% Pb/C (16 mg, 0.015 mmol) and 1,4-cyclohexadiene (141 μL, 1.5 mmol) were added. The solution was stirred reflux for 5 h. The resulting solution was concentrated after filtrating and purified by column chromatography (DCM/MeOH = 40:1) to give **7p** as yellow solid (11 mg, 61.4%).

*(E)-2-((2-(3,4-dihydroxyphenyl)-5-hydroxy-4-oxo-4H-chromen-7-yl)oxy)ethyl 3-(4-acetoxyphenyl)acrylate* (**7q**): To a solution of compound **28q** (100 mg, 0.15 mmol) in DCM (5 mL), CF_3_COOH (0.55 mL, 7.5 mmol) was added. The solution was stirred at room temperature for 12 h. The resulting solution was concentrated and purified by column chromatography (DCM/MeOH = 30:1) to give **7q** as yellow solid (16.7 mg, 31.4%).

### 3.2. In Vitro BACE1 Enzyme Assay

The BACE1 FRET assay kit was purchased from the PanVera Co. (Invitrogen, USA). The assay was carried out according to the supplied manual with modifications. Briefly, assays were performed in triplicate in 384-well black plates with a mixture of 5 μL of BACE1 (1.0 U/mL), 5 μL of the substrate (750 nM, Rh-EVNLDAEFK-Quencherin 50 mM, ammonium bicarbonate), and 5 μL of compound dissolved in 10% DMSO. The fluorescence intensity was measured with a TECAN infinite 200 microplate reader for 60 min at 25 °C in the dark. The mixture was irradiated at 544 nm and the emission intensity recorded at 590 nm. The percent inhibition (%) was obtained by the following equation: Inhibition % = (1 −SS/SC) × 100%, where SC is the slope of fluorescence change of the control (enzyme, buffer, and substrate) during 60 min, and SS is the slope of fluorescence change of the tested samples (enzyme, sample solution, and substrate) during 60 min of measurement. IC_50_ values were calculated from the nonlinear curve fitting of percentage inhibition against inhibitor concentration using Prism 3.0 software (GraphPad Software, San Diego, CA, USA).

### 3.3. Inhibition of Aβ Production in APP Overexpressed Cells

HEK293 cells stably overexpressing human APP695 were plated in a poly-d-lysine-coated 96-well plates and grown to a density of 80–90%. Compounds were dissolved in DMSO to make a 10 mM stock solution and diluted in DMEM complete medium to the final concentrations: 0.1, 1.0, 5.0, 10.0, 25.0, and 50.0 µM. DMSO was used at a 1:200 dilution as a negative control and a commercially available BACE1 inhibitor (C3, also known as BACE inhibitor IV) as a positive control. After 23 h of incubation, the cell supernatants were directly used for measurement in the V-Plex Aβ Peptide Panel 1 (6E10) Kit from Meso Scale Diagnostics (Rockville, MD, USA) for measuring Aβ38, −40, and −42 simultaneously. Cells were washed with PBS and lysed in Triton X100 lysis buffer for subsequent protein concentration (BCA) measurement.

### 3.4. Kinetic Analysis

The Dixon plot is a graphical method (plot of 1/enzyme velocity (1/V) against inhibitor concentration (I)) for determination of the type of enzyme inhibition, which was used to determine the dissociation or inhibition constant (Ki) for the enzyme inhibitor complex [17]. The Dixon plots for BACE1 inhibition were obtained in the presence of various concentrations of BACE1 substrate (250, 500, 750 nM) and the concentrations of **4l**, **7c**, and **7k** as follows: 0.1 µM, 0.5 µM, 1.0 µM. In this way, the inhibition constants (Ki) of **4e** were determined by interpretation of Dixon plots, where the value of the *x*-axis represents −Ki when 1/V = 0.

### 3.5. MTT Assays

The cytotoxicity was examined by using 3-[4,5-dimethylthiazol-2-yl]-2,5-diphenyl-tetrazolium bromide (MTT) assay. HEK293T cells were seeded with complete DMEM into 96-well plates (104/well, 100 L/well), incubated at 37 °C in a humidified atmosphere with 5% CO_2_ for 24 h. All the compounds were diluted with the phenol red-free DMEM at a specific concentration. After 24 h of exposure to different concentrations of compound, the supernatant was removed and then 20% (*v/v*) MTT (5 mg/mL) diluted with fresh culture medium was added into each well and incubated for 4 h. Cells were lysed with 100 μL dimethyl sulfoxide (DMSO) and shaken for 10 min at room temperature. The absorbance in each well was detected at a wavelength of 490 nm using a Universal Microplate Reader (ELX 800 lv, BIO-TEK Instruments, Inc., Winooski, VT, USA).

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
