# Peer review of "Extensive Structure Modification on Luteolin-Cinnamic Acid Conjugates Leading to BACE1 Inhibitors with Optimal Pharmacological Properties"

_molecules, 2019, doi:10.3390/molecules25010102_

Round 1

Reviewer 1 Report

In this manuscript, the authors reported the synthesis of luteolin-cinnamic acid conjugates and evaluation of their inhibitory activity against BACE1 on molecular and cell level. The Topic is interesting, but,

In introduction section, the reasons for the design of the targeted compounds should be presented clearly, for example, why luteolin and cinnamic acid were selected as building block for BACE1 inhibition, and what is the reason for linker design. The results and discussion section should be reorganized, the chemical synthesis and biological evaluation should be separated. The synthesis of compounds 4a-4i cannot be found. Why the inhibition of some compounds (5f, 5c, 7a-7d, 7k-7l) at 10 uM is ~ 60%, while, their IC50 vary from 0.23 to 28 uM? Please clarify. I cannot see any dose-dependence from the results in Figure 6, Please use some words carefully. Figure 7, the legend is not clear.

Overall, the current version cannot be considered for publication in Molecules.

Author Response

Point 1:In introduction section, the reasons for the design of the targeted compounds should be presented clearly, for example, why luteolin and cinnamic acid were selected as building block for BACE1 inhibition, and what is the reason for linker design.

Response 1:In our previous work (ref 13), luteolin(1) and p-hydroxy-cinnamic acid (2) were found to simultaneously bind different part of BACE1 protein by STD-NMR experiment. We designed the conjugates to linker two segments through proper linkage so as to improve the potency.. Conjugate 3, which was found in our previous work as the optimal inhibitor, was further modified in this paper..

We have modified our manuscript (line 55-60) to explain our design concisely.

Point 2:The results and discussion section should be reorganized, the chemical synthesis and biological evaluation should be separated.

Response 2:Well, although the reviewer’s suggestion is in line with the organization of many papers, we intend to keep its current style because the evolution of three series compounds depict the process of our structural optimization more clearly. If we split the synthesis and bioassay of a series of compounds, it may be more difficult to understand the logic and process of our work.

Point 3:The synthesis of compounds 4a-4i cannot be found.

Response 3: Such information was shown detailed in the Supplementary Materials except the synthesis of compound 4j.

Point 4:Why the inhibition of some compounds (5f, 5c, 7a-7d, 7k-7l) at 10 uM is ~ 60%, while, their IC50 vary from 0.23 to 28 uM? Please clarify.

Response 4:The clarification could be shown at Line 173-177, and a S-curve for IC50 calculation of several compounds was added in Supplementary Materials. This may hint the mechanism for their inhibition is somewhat different. However, the further investigation of this difference is not the topic of this paper.

Point 5:I cannot see any dose-dependence from the results in Figure 6, Please use some words carefully.

Response 5:We rephrase the sentence as “At the higher concentrations (10, 25 and 50 µM) 7b-d and 1 dose-dependently reduced Aβ levels.” (line 211)

Point 6:Figure 7, the legend is not clear.

Response 6: A brief legend was added at Line 221-224

Reviewer 2 Report

This manuscript describes the investigation of the inhibitory effects on the enzyme beta-secretase 1 (BACE1) by synthesized conjugates derived from luteolin and p-hydroxy-cinnamic acid. BACE1 is a key enzyme in the generation of amyloid-β (Aβ) peptides in the neurons, which are accumulated in Alzheimer patients and thus it is believed that compounds that are able to slow down the formation of for example Aβ may be one approach to develop new drugs against Alzheimer’s disease. The overall idea with the work described in this manuscript is highly relevant as only a few drugs are on the marked for this deadly neurodegenerative disease. The present manuscript is focused and well structured, and the synthetic work is well performed and described in such a detail that other researchers are able to reproduce the synthesis of the different conjugates. However, the experimental part that also includes NMR and mass spectrometric data is very comprehensive. I therefore suggest that the NMR and mass spectrometric data are included in Supplementary Materials as these data are not essential for understanding the content of the manuscript, although these data are highly important and necessary for the documentation of the identity of the synthesized conjugates. With regard to the mass spectrometric data only the quasi-molecular ions are shown (please use brackets instead of parenthesis for to indicate ions) but should also include major fragment ions of the synthesized conjugates. Furthermore, an exact mass of the different synthesized conjugates needs to be determined in order to verify their chemical composition.

Finally, the bioavailability of the tested conjugates needs to be discussed in the manuscript. The synthesized conjugates are tested in vitro for enzyme inhibition and cytotoxicity, and SAR studies are performed which is fine. However, potential drug compounds targeting the brain, which are active in vitro may not necessary be active in vivo as they have to cross the blood–brain barrier (BBB) that is a highly selective semipermeable border. The BBB often represent a challenge in the development of drugs being active in the brain. The authors therefore need to address this challenge because the conjugates that are described in this manuscript may have a problem crossing the BBB.

In conclusion, I find that this is a fine manuscript that contain a comprehensive piece of work but need revision as indicated above before it is acceptable for publication in Molecules.

Author Response

Point 1&2:However, the experimental part that also includes NMR and mass spectrometric data is very comprehensive. I therefore suggest that the NMR and mass spectrometric data are included in Supplementary Materials as these data are not essential for understanding the content of the manuscript, although these data are highly important and necessary for the documentation of the identity of the synthesized conjugates. With regard to the mass spectrometric data only the quasi-molecular ions are shown (please use brackets instead of parenthesis for to indicate ions) but should also include major fragment ions of the synthesized conjugates. Furthermore, an exact mass of the different synthesized conjugates needs to be determined in order to verify their chemical composition.

Response 1:The data had been moved to the Supplementary Materials part.

Response 2:The HR-MS data of target compounds were added to verify their chemical composition.

Point 3:Finally, the bioavailability of the tested conjugates needs to be discussed in the manuscript. The synthesized conjugates are tested in vitro for enzyme inhibition and cytotoxicity, and SAR studies are performed which is fine. However, potential drug compounds targeting the brain, which are active in vitro may not necessary be active in vivo as they have to cross the blood–brain barrier (BBB) that is a highly selective semipermeable border. The BBB often represent a challenge in the development of drugs being active in the brain. The authors therefore need to address this challenge because the conjugates that are described in this manuscript may have a problem crossing the BBB.

Response 3: We calculate cLogP and BBB level of some representative conjugates, and discuss the results in the context and (line 230– 233). The software info can be seen in the Supplementary Materials.

Reviewer 3 Report

in this paper authors aimed at investigating thge effect of a BACE 1 inhibitor conjugated with luteolin and cinnamic acid. They found out improved pharmacological effects with a weaker cytotoxicity. 

I have one question for authors: do they think that conjugating different molecules (not just BACE 1 inhibitors) with luteolin or cinnamic acid could improve the pharmacodynamics of also other chemical compounds active in APP processing or is this just specific to BACE 1 inhibitors?? 

Author Response

Point 1: I have one question for authors: do they think that conjugating different molecules (not just BACE 1 inhibitors) with luteolin or cinnamic acid could improve the pharmacodynamics of also other chemical compounds active in APP processing or is this just specific to BACE 1 inhibitors?? 

Response 1: Maybe the conjugates not only directly target BACE1 as BACE1 inhibitors, but also influence APP processing by other ways. We have made a preliminary observation on the activity of the conjugate other than BACE1 inhibition, but the result is not released here.

Luteolin and related flavones have been known to exhibit multiple activities, so the potential of conjugates composed of luteolin (flavone) and other molecules may not limit to BACE1 inhibition. In our opinion, multi-target mode modulation of APP processing of the conjugates can be beneficial to their AD therapeutic potential.

Round 2

Reviewer 1 Report

The revised manuscript addressed the reviewers' concerns, can be accepted in present form.